# Modification of *Artichoke* Dietary Fiber by Superfine Grinding and High-Pressure Homogenization and Its Protection against Cadmium Poisoning in Rats

**DOI:** 10.3390/foods11121716

**Published:** 2022-06-12

**Authors:** Renwei Zhu, Tianhui Xu, Bian He, Yayi Wang, Linwei Zhang, Liang Huang

**Affiliations:** 1School of Food Science and Engineering, Central South University of Forestry and Technology, Changsha 410004, China; zhurenwei@126.com (R.Z.); 1886486651@163.com (T.X.); 2222@163.com (B.H.); yywang97@126.com (Y.W.); 20211100410@csuft.edu.cn (L.Z.); 2School of Materials and Chemical Engineering, Tongren University, Tongren 554300, China; 3Hunan Key Laboratory of Processed Food for Special Medical Purpose, Changsha 410004, China

**Keywords:** compound modification, *artichoke* dietary fiber, physicochemical properties, cadmium poisoning

## Abstract

This study was carried out to investigate the effects of superfine grinding (SP) and high-pressure homogenization (HPH) on the structural and physicochemical properties of *artichoke* dietary fiber (ADF), as well as the protective effects against cadmium poisoning in rats. The structural characteristics and physicochemical properties of ADF, HPH-ADF (ADF treated by HPH) and CM-ADF (ADF treated by SP and HPH) were determined, and cadmium chloride (CdCl_2_) was induced by exposing rats for 7 weeks. The amounts of creatinine and urea; the activities of aspartate aminotransferase (AST) and alanine aminotransferase (ALT) in serum; the quantity of red blood cells, hemoglobin, white blood cells and neutrophil proportion in blood samples; and the activity of glutathione peroxidase (GSH-Px) in liver tissue were analyzed. Hematoxylin-eosin (HE) staining was performed to analyze the tissue structure and pathology of the liver and testis. The results showed that ADF subjected to HPH and SP-HPH exhibited increased content of soluble dietary fiber (SDF) (*p* < 0.05). HPH and SP-HPH treatments increased oil-holding capacity (OHC), total negative charge (TNC) and heavy metal adsorption capacity (*p* < 0.05). The CdCl_2_ intervention led to a significant increase in AST, ALT, creatinine, urea, neutrophil proportion and white blood cell count, as well as a significant decrease in GSH-Px activity, red blood cell count and hemoglobin (HGB) (*p* < 0.05). In rats fed with ADF, HPH-ADF and CM-ADF significantly reduced creatinine, urea amounts, ALT, AST activity in serum, leukocyte count and the neutrophil ratio in blood and increased GSH-Px activity in the liver, in addition to increasing the erythrocyte count and hemoglobin count in blood (*p* < 0.05). H&E staining results showed that steatosis in the liver was significantly reduced, whereas testicular tissue edema was improved. These results indicate that ADF exhibited positive activity against cadmium poisoning in rats and that CM-ADF had a better protective effect than ADF and HPH-ADF. ADF has specific potential to be used in health foods or therapeutic drugs, providing a reference for the development and utilization of artichoke waste.

## 1. Introduction

*Artichoke*, a vegetable native to the Mediterranean coast of Europe, is an important ingredient of the Mediterranean diet containing a high content of fructans and other dietary fibers [1]. The edible flowerheads of *artichoke* are used in the agrifood industry for the preparation of fresh, canned and frozen products [2], which are highly appreciated by consumers for their organoleptic properties and nutritional value [3]. The *artichoke* processing industry generates large amounts of solid waste consisting mainly of stems and bracts, the external parts of the flowers. The ratio of edible fraction/total biomass is very low (15–20%), with the discarded parts representing 70–80% of the total *artichoke* flower [4]. *Artichoke* cultivation in China can be traced back to 2005, when it was introduced and planted in Changde, Hunan Province, China [5]. Through constant exploration and experiments, researchers have conquered the key technologies to limit the seedling, tissue culture and pest control of *artichoke*, with the annual output of the country reaching 100,000 tons. With the annual increase in production, waste is produced, such as *artichoke* buds, which also shows explosive growth; therefore, research on the renewable utilization of *artichoke* waste has become more urgent [6]. *Artichoke* waste is rich in dietary fiber; however, there is a scarcity of literature focused on discarded *artichoke* bracts as a source of dietary fiber. The major components of *artichoke* dietary fiber (ADF) are insoluble, whereas soluble dietary fiber (SDF) is of greater benefit to human health than insoluble dietary fiber (IDF) [7]. Dietary fiber has important physiological functions, such as prevention and treatment of rectal cancer [8], prevention and improvement of cardiovascular diseases, regulation of blood sugar levels [9], improvement of intestinal flora [10] and adsorption of heavy metals. Therefore, modification of IDF to increase SDF has become an important research direction for the improvement of the functional properties of ADF in food industry.

Modification of dietary fiber can be achieved by physical, chemical and biological methods. High-pressure homogenization (HPH) is a novel and non-thermal processing technology that has proven to be an effective technology in the food industry [11,12], especially for modification of dietary fiber [13]. Spotti et al. [14] found that HPH-treated dietary fiber had a higher ratio of soluble fraction than HPH treated by high hydrostatic pressure at 200 MPa, with improved physicochemical and functional properties. Hua et al. [15] found that HPH could convert about 8% of the insoluble fibers to soluble fibers and modify the microstructure of fibers. Superfine grinding (SP), i.e., increasing SDF content by reducing IDF, is a form of dry comminution. Furthermore, SP was reported to significantly increase various physicochemical properties (e.g., water-holding capacity, swelling capacity, oil-holding capacity and cation exchange capacity), glucose adsorption capacity, α-amylase inhibitory activity and pancreatic lipase inhibitory activity of IDF to varying extents [16,17]. However, there is little scientific information available on the influences of combination treatment with SP and HPH on the structure and functional properties of ADF. In this study, we combined SP with HPH to modify ADF and improve its functional properties. We also treated ADF with HPH to better evaluate the effects of SP-HPH treatment on ADF.

Cadmium poisoning refers to the poisoning reaction caused by the human body inhaling cadmium dust or eating high-cadmium food over a long period of time and can be divided into three types: chronic cadmium poisoning, acute cadmium poisoning by inhalation and acute cadmium poisoning by ingestion. Cadmium is extremely harmful to the human body; long-term excessive exposure causes kidney damage. Kidney damage induced by chronic cadmium poisoning is mainly manifested as renal tubular reabsorption disorders, which can appear as hyperphosphaturia, hyperaminoaciduria and glucosuria. It can also lead to progressive obstructive emphysema, pulmonary fibrosis, bone damage and other diseases. Acute cadmium poisoning can lead to nausea, diarrhea, muscle aches and even acute renal failure and death within a few hours. To date, no specific treatments have been developed for chronic cadmium poisoning, and only a limited number of treatments for acute cadmium poisoning are available. For example, chelator methods, such as with dimercaptosuccinic acid and ethylenediaminetetraacetic acid, whereby the chelate binds to Cd^2+^ and eventually removes Cd^2+^ through the kidneys. However, this method can have side effects on the body and is deficient in terms of safety and efficacy. Therefore, safe and low-cost dietary therapy has attracted the attention of scholars. Many studies have found that plants with a large amount of fiber are able to bind heavy metal ions. The -COOH and -OH groups of cell wall polysaccharides and proteins are considered to be the main functional groups that bind heavy metal ions, and this phenomenon is also manifested in dietary fibers [18].

In the present study, the effects of HPH and SP-HPH on the structure and physical and chemical properties of ADF were studied. Furthermore, we explored methods to improve the physical and chemical properties of ADF, as well as the protective effects against cadmium poisoning in rats, which lays a foundation for research and development of ADF and increased recycling of discarded *artichoke* bracts.

## 2. Materials and Methods

### 2.1. Materials and Reagents

*Artichoke* waste was obtained as a byproduct of canned *artichoke* processing from Huimei Agricultural Technology Co., Ltd. (Changde, China). Alkaline protease (enzyme activity ≥ 200 U/mg), high-temperature- resistant α-amylase (enzyme activity ≥ 40 U/mg) and diastatic enzyme (enzyme activity ≥ 10 U/mg) were obtained from Shanghai Yuanye Biotechnology Co., Ltd. (Shanghai, China). All other chemicals were of analytical grade. Basal diet (21% casein, 0% ADF, 7% sunflower oil, 1% cholesterol, 15% sucrose, 51.2% starch, 0.3% methionine, 3.6% minerals, 1% mixed vitamins), ADF special diet (basal diet plus 2% ADF), HPH-ADF special diet (basal diet plus 2% HPH-ADF) and CM-ADF special diet (basal diet plus 2% CM-ADF) were prepared by Jiangsu Medison Biomedical Co. (Jiangsu, China). The kits for measurement of glutathione peroxidase (GSH-PX) were obtained from Nanjing Jiancheng Bioengineering Institute (Nanjing, China). The kits for measurement of aspartate aminotransferase (AST), alanine aminotransferase (ALT), creatinine (CREA-S) and urea were obtained from Shenzhen Mindray Bio-medical Electronics Co. Ltd. (Shenzhen, China). Cadmium chloride (CdCl_2_) of analytical grade was supplied by Sinopharm Chemical Reagent Co. Ltd. (Shanghai, China).

### 2.2. Sample Preparation

ADF was prepared by the method of Association of Official Analytical Chemists (AOAC 991.43) with slight modifications. Briefly, 10 g *artichoke* waste powder and 100 mL water were treated with heat-stable α-amylase (250 mg/L) at 95 °C and pH 5.5 for 1 h. The suspension was cooled to 60 °C and incubated with neutral protease (3000 mg/L) at 55 °C and pH 8.0 for 2 h. Next, the suspension was digested with amyloglucosidase (350 mg/L) at 60 °C and pH 4.2 for 1 h. Following enzyme hydrolysis, the suspension was heated in a boiling water bath for 10 min to inactive all the enzymes. Then, 95% ethanol (4 times the volume of the hydrolysate) was added to precipitate polysaccharides and left overnight. The residue was collected and oven-dried to obtain the ADF used for further study. The ADF was crushed by an air-induced grinder to obtain ADF (crushing time: 10 min). HPH-ADF was obtained by low-temperature freeze drying after homogenization by a high-pressure homogenizer (homogenization pressure, 97 MPa; homogenization temperature, 41 °C; material concentration, 2.5%). CM-ADF (material ratio, ADF: zirconium ball = 1:5; homogenizing pressure, 97 MPa; homogenizing temperature, 41 °C; material concentration, 2.5%; homogenizing time, 8 h) was obtained after composite modification with a high-energy nanoball mill and high-pressure homogenization.

### 2.3. Chemical Analysis and Determination of Particle Size

Total dietary fiber (TDF), as well as IDF and SDF contents, was determined by AOAC (2000) methods. The sample particle size was measured by a micron particle size meter LS13320 (Beckman Coulter, Inc. Brea, CA, USA), and air was used as dispersant. The sample was dispersed evenly with ultrasonic assistance. The refractive index was set as 1.530, the medium refractive index was 1.42 and the particle size was measured from 0.01 μm to 2500 μm.

### 2.4. Scanning Electron Microscopy

The sample structure was measured by an FEG-250 scanning electron microscope (FEI Company, Hillsboro, OR, USA). Appropriate samples were placed in the sample seat, and the height of the sample was adjusted so that it was not higher than the edge of the sample seat. The acceleration voltage was set at 5 kV, and the working distance was set at 10 mm. The sample particle morphology was observed at 5000 times and 40,000 times magnification, and the changes in sample structure were analyzed by photography.

### 2.5. Fourier Transform Infrared Spectroscopy (FTIR)

FTIR analysis was performed with an FTIR IRTracer-100 (SHIMADZU Corporation, Kyoto, Japan) from 400 to 4000 cm^−1^ with 32 scans and 4 cm^−1^ resolution. A dried sample (0.4 mg) was ground with KBr (40 mg) in an agate mortar, sheeted to one slice and scanned with a blank KBr background.

### 2.6. Determination of Water-Holding Capacity (WHC)

The water-holding capacity (WHC) was determined in triplicate according to the method described by Xie et al. [19] with minor modifications. Briefly, 100 mL of distilled water was mixed with 10 g of sample and oscillated at room temperature (25 °C) for 2 h. After centrifugation at 3500× *g* rpm for 15 min, the sediment was collected and weighed. The WHC was calculated by Equation (1):(1)$$WHC(g/g)=\frac{W_{1}-W}{W}$$
where *W*_1_ is the wet weight, and *W* is the dry weight.

### 2.7. Determination of Oil-Holding Capacity (OHC)

The oil-holding capacity (OHC) was determined in triplicate according to the method described by Zhang et al. [20] with slight modifications. Briefly, 40 mL of peanut oil was mixed with 10 g of sample at 4 °C for 2 h. Then, the mixture was centrifuged at 3500× *g* rpm for 15 min, and the sediment was collected and weighted. The OHC was calculated by Equation (2):(2)$$OHC(g/g)=\frac{W_{1}-W}{W}$$
where *W*_1_ is the weight of the residue (g) containing oil, and *W* is the weight of the residue (g) containing oil.

### 2.8. Total Negative Charge (TNC)

The total negative charge (TNC) was determined in triplicate according to the method described by Marshall et al. [21] with minor modifications. Briefly, 2 g of the fully protonated sample was dissolved in 50 mL, 0.1 mol/L sodium hydroxide solution and stirred at room temperature for 24 h; then, filtrate was extracted. Then, 10 mL of filtrate was added to 15 mL, 0.1 mol/L hydrochloric acid solution and titrated with 0.1 mol/L sodium hydroxide solution to the end point after full reaction, the total negative charge was calculated.

### 2.9. Determination of Heavy Metal Adsorption Capacity

The heavy metal adsorption capacity was determined in triplicate according to the method described by Wang et al. [22] with slight modifications. Briefly, 5.0 g of sample was suspended in a solution (100 mL) containing 1.0 mg/mL each of CuSO_4_, CdCl_2_, Pb(NO_3_)_2_ and Hg(NO_3_)_2_ in a 250 mL conic flask, and the pH was adjusted to 2.0 and 7.0. The slurries were shaken for 36 h at room temperature. Following adsorption, 2 mL volume of the sample was collected, and absolute ethanol (8 mL) was added to precipitate the SDF. The mixture was centrifuged at 4200× *g* rpm for 20 min. The concentrations of Cu^2+^, Cd^2+^, Pb^2+^ and Hg^2+^ in the supernatant were determined by atomic absorption spectrometry (Shimazu AA-6300, Shimadzu, Kyoto, Japan).

### 2.10. Animals and Experimental Method

SD male rats (33, specific-pathogen-free, SPF) weighing about 250 g each were provided by Hunan Ansheng Mei Pharmaceutical Research Institute Co., Ltd. All animals were adaptively fed in an environment with controlled temperature (24 ± 2 °C) and light (12 h of light and 12 h of dark) for one week before the start of the experiment (animal production license number: SYXK (Xiang) 2018-0004 (Changsha, China)). All experimental procedures and facilities were approved by the Laboratory Animal Welfare Ethics Committee of Hunan Ansheng Pharmaceutical Research Institute Co., Ltd. The project identification code was IACUC-2020038,and the approval date was 13 July 2020. A 50 mg/L aqueous solution of cadmium chloride was prepared with cadmium chloride and deionized water. Rats were randomly divided into five groups, as shown in Table 1: blank group (BG), control group (CG), ADF group, HPH-ADF group and CM-ADF group. The rats were fed continuously for 7 weeks according to the above groupings. During the experiment, rats were observed daily for appetite, coat and mental status. All rats were fasted from food and water for 12 h at the end of 7 weeks and anesthetized by intraperitoneal injection of chloral hydrate and rapidly deprogrammed. Following anesthesia, blood was immediately collected from the abdominal aorta, liver and testes were dissected, and some of the liver and testes were fixed in paraformaldehyde and stored on dry ice for the determination of glutathione peroxidase activity and other indicators, and the rest were stored in a refrigerator at −80 °C for further analysis.

### 2.11. Determination of Plasma Transaminase (AST and ALT) Activities, CREA-S, UREA, Red Blood Cell Count, Hemoglobin Count, White Blood Cell Count and Neutrophil Ratio Levels

Intravenously obtained blood samples of the rats were allowed to coagulate before centrifuging at 2000× *g* rpm for 25 min. The serum was collected, and the activities of AST, ALT, CREA-S and urea were determined with corresponding commercial kits (Mindray, Shenzhen, China). Red blood cell count, hemoglobin count, white blood cell count and neutrophil ratio of whole blood were measured with a BS-430 automatic biochemical analyzer (Beijing Plantronics Co., Beijing, China).

### 2.12. Determination of Glutathione Peroxidase (GSH-PX) Activity in Liver

About 0.5 g of liver tissue was rinsed with cold physiological saline at 5 °C to 10 °C. The blood on the surface of the liver was removed and carefully dried with absorbent paper. The liver was ground into liver homogenate with a tissue grinder, and the prepared homogenate was centrifuged at 3500× *g* r/min at low temperature for 18 min. The supernatant was collected and determined according to the procedures of the GSH-PX kit.

### 2.13. Histology and Case Analysis of Liver and Testis in Rats

For histopathological analysis, liver and testis tissues were fixed in 9% paraformaldehyde for 72 h, embedded in paraffin for the preparation of tissue sections (5 μm thickness) and stained with hematoxylin and eosin (HE).

### 2.14. Statistical Analysis

All experiments were conducted in triplicate, except for five or seven rats in animal experiment. Data were analyzed in triplicate by one-way analysis of variance using the SPSS 20.0 software package for Windows and reported as mean ± standard deviation (SD). *p* < 0.01 was considered an extremely significant difference. *p* < 0.05 was considered a significant difference.

## 3. Results and Discussion

### 3.1. Chemical Analysis and Determination of ADF Particle Size

The chemical compositions of ADF subjected to different treatments are shown in Table 2. Following the two treatments with dietary fibers (HPH-ADF, 15.06 g/100 g; CM-ADF, 20.51 g/100 g) SDF contents were increased as compared with the control (ADF, 6.45 g/100 g). Moreover, the increase in SDF content (13.97 g/100 g) was more evident for the SP-HPH treatment. In addition, the two treatments deceased IDF contents in dietary fibers (HPH-ADF, 70.86 g/100 g; CM-ADF, 65.58 g/100 g) as compared with the control (ADF, 79.73 g/100 g). No significant difference (*p* > 0.05) was observed in terms of total dietary fiber contents (ADF, 86.18 ± 1.33 g/100 g; HPH-ADF, 85.92 ± 1.25 g/100 g; CM-ADF, 86.09 ± 0.77 g/100 g). The particle size distributions of ADF, HPH-ADF and CM-ADF are displayed in Figure 1. The results indicate that SP and HPH treatment reduced the particle size of ADF. Zhu et al. [23] reported that multidimensional swing high-energy nanoball milling reduced the size of dietary fiber particles to the submicron scale. The main cause of the breakup of particles in the HPH process was the shear stress encountered by fluid, as previously reported by Clarke et al. [24]. It is worth noting that HPH-SP combined treatment further reduced the particle size of ADF in comparison to treatment with HPH alone. Moreover, the particle size of CM-ADF presented a normal distribution, with the peak located at about 15 μm. This may be due to the fact that after ultrafine grinding, the internal structure of ADF was fully opened, and then after high-pressure homogenization, the particles of CM-ADF were further broken down in a high-pressure and high-temperature environment, and the particle size was further reduced and distributed more evenly [25].

### 3.2. Scanning Electron Microscopy

The particle morphology of ADF subjected to different treatments was observed by SEM; results are displayed in Figure 2. The particle sizes of HPH-ADF and CM-ADF were all smaller than ADF (Figure 2A). This is consistent with the results reported by Yan et al. [26] and Ulbrich et al. [27]. Furthermore, the particles of CM-ADF were clearer and more uniform than those of ADF and HPH-ADF, which might be helpful in absorbing water. Thus, the loose fiber structure with small particles and multiple holes was attributed to the combination effect of SP and HPH treatment (Figure 2B).

### 3.3. FTIR Spectra

Figure 3 shows the FTIR spectra of ADF, HPH-ADF and CM-ADF. The samples of all four groups had strong signals at 3400 cm^−1^, which indicates that the vibrations of the hydrogen bond were mainly provided by the hydroxyl groups of cellulose and hemicellulose [14]. The peaks at 2923–2927 cm^−1^ were attributed to the C-H stretching bands of the methylene group of polysaccharides [28]. The peak near 1735 cm^−1^ was due to vibrations of acetyl and uronic ester groups of hemicelluloses or ester linkage of carboxylic groups of the ferulic and p-coumaric acids of hemicelluloses [29]. Most hemicelluloses had been dissolved after SP and HPH treatment, which could explain the fact that the peaks at 1735 cm^−^^1^ for in the CM-ADF group were significantly weaker than those of the ADF and HPH-ADF groups. The absorption peak at 1629–1639 cm^−1^ was due to an H-O-H bending vibration peak of adsorbed water, and the peak at 1417–1421 cm^−1^ was attributed to a CH_2_ bending vibration peak of cellulose [30]. The peak near 1375 cm^−1^ belonged to the aliphatic asymmetric C-H bond bending vibration peak, the absorption peak at 1321–1324 cm^−1^ belonged to the carbohydrate CH_2_ bending vibration peak, and the peak at 1047–1053 cm^−1^ belonged to the C-O stretching vibration peak in carbohydrate [31]. The peak at 604–617 cm^−1^ was associated with the in-plane swing of C-H. No characteristic absorption peak of lignin appeared near 1530 cm^−1^, indicating that none of the four groups of samples contained lignin. The results showed that *artichoke* dietary fiber had special absorption peaks, such as C-H bond, C=O bond, H-O-H bond, CH_2_ bond and C-O bond, and no new absorption peak appeared with modification. Thus, modification had no effect on the chemical composition of *artichoke* dietary fiber. Compared with ADF and HPH-ADF, the peak strength of alkyl and oxygen-containing functional groups of CM-ADF gradually increased, which might indicate that the composite modification method was better than the single modification method in terms of the conversion rate of IDF to SDF.

### 3.4. WHC, OHC and TNC

The WHC, OHC and TNC of ADF, HPH-ADF and CM-ADF are shown in Table 3. The HPH and SP-HPH treatments reduced the water-holding capacity of ADF. This is consistent with the research results reported by Xie et al. [14], Chau et al. [32] and Zhu et al. [16]. Analysis indicates that IDF was modified into SDF after ADF was modified by HPH and SP, which resulted in a decrease in the determination result of water-holding capacity. To verify this problem, the content of SDF in the supernatant was determined by the AOAC method (2000); the result is shown in Table 3. The SDF content of CM-ADF was the highest, followed by HPH-ADF and ADF.

Three modification methods were found to significantly improve the OHC of ADF. Zhang et al. [33] reported that HPH treatment increased the OHC of citrus fibers relative to unmodified fibers, and Zhu et al. [34] found that the OHC of grape pomace dietary fiber increased by 1.5 times after superfine grinding. However, the OHC of CM-ADF was 3.25 ± 0.46 g/g, which was higher than that of HPH-ADF (*p* < 0.05). The surface of CM-ADF was more porous and had larger surface area than those of ADF and HPH-ADF, which was proven in Section 3.2 of this paper. Thus, CM-ADF could absorb more oil, reducing the absorption of oil by the human body, as well as cholesterol content.

The total negative charge of CM-ADF was much higher than that of ADF and HPH-ADF, and the metal cation adsorption capacity of CM-ADF was significantly enhanced (*p* < 0.05), possibly due to the change in functional group strength of ADF after the SP-HPH treatment, which led to the change in total negative charge. This effect can likely be attributed to the synergistic disrupting effect of SP and HPH on the structure of ADF, exposing more negatively charged functional groups [35].

### 3.5. Determination of Heavy Metal Adsorption Capacity

Table 4 shows the adsorption capacity of ADF, HPH-ADF and CM-ADF for Cd (II), Cu (II), Pb (II) and Hg (II). The adsorption effect of ADF on heavy metal ions at pH 7 was significantly stronger than that at pH 2 (*p* < 0.05). This is consistent with the research results reported by Zhang et al. [36]. These results indicate that the adsorption site of ADF to heavy metals was mainly located in the intestine rather than the stomach. Compared to ADF, HPH and SP-HPH treatments significantly increased the binding capacity of heavy metal ions. In particular, the heavy metal adsorption capacity of CM-ADF at pH 7 was as follows: Cd (II): 29.96 ± 0.46 mmol/g, Cu (II): 25.37 ± 0.13 mmol/g, Pb (II): 44.19 ± 0.22 mmol/g, Hg (II): 26.77 ± 0.12 mmol/g, all of which were higher than that of HPH-ADF and ADF (*p* < 0.05).

### 3.6. Effects of Artichoke Dietary Fiber on Serum AST and ALT Levels, Creatinine, Urea and Blood Routine Indices in Rats

The serum AST and ALT, as well as creatinine and urea levels, in rats are shown in Table 5. Compared with the BG group, serum creatinine, urea, AST and ALT increased by 148%, 126.4%, 153.2% and 186.3%, respectively, in the CG group, indicating that Cd^2+^ caused serious liver cell membrane and kidney damage in rats during the feeding period. Serum creatinine, urea, AST and ALT levels of rats in the ADF group, HPH-ADF group and CM-ADF group decreased by 14.6%, 28.1%, 41.1%, 16.7%, 31.7%, 45.8%, 9.1%, 24.0%, 47.4%, and 14.1%, 31.1%, 40.2%, respectively, compared with the CG group. The CM-ADF group showed the most significant downward trend, indicating that ADF had a certain protective effect on the liver cell membrane and kidney of rats and that ADF modified by HPH-SP had the best protective effect, which may be due to the increase in the number of oxygen-containing functional groups and the high cation adsorption capacity of ADF, which could effectively remove Cd^2+^ from the rat’s body after compound modification. However, the blood creatinine and blood urea levels of rats in the ADF intervention group were higher than those in the BG group, indicating that ADF only played a limited protective role and could not completely prevent the kidney damage caused by Cd^2+^. Dietary fiber effectively reduced serum creatinine levels, as reported by Koguchi et al. [37] and Chiavaroli et al. [38].

The blood routine indices in rats are depicted in Table 6. The CG group erythrocyte count was 20% lower than that of BG group rats. The CG group white blood cell count and neutrophil proportion were 91.20% and 43.81% higher than those of BG rats, respectively. Cd^2+^ remained in the rat intestine after being absorbed into the blood. Blood transport during the combination treatment, as well as red blood cell membrane and plasma albumin and Cd^2+^, stimulates metallothionein and reactive oxygen species, causing a red blood cell and oxidative stress response of T and B lymphocytes, resulting in a reduced red blood cell count and an increased white blood cell count and neutrophil ratio. The erythrocyte count of rats in the ADF group, HPH-ADF group and CM-ADF group increased by 10.96%, 10.83% and 10.71%, respectively, compared with the CG group. The ratio of white blood cell count and neutrophil in the ADF group, HPH-ADF group and CM-ADF group decreased by 8.49%, 40.60%, 40.60% and 4.11%, 7.53% and 12.33%, respectively, compared with the CG group.

### 3.7. Effects of ADF on GSH-PX Activity in Rat Liver

The effect of ADF on GSH-PX activity in rat liver is shown in Figure 4. The GSH-PX activity of the CG group was 160% lower than that of BG group rats, indicating that Cd^2+^ seriously damaged the reducing substance in the rat liver during feeding, resulting in significantly reduced GSH-PX activity in rat liver, as well as impaired liver function. Compared with CG group rats, GSH-PX activity in the liver of rats in the ADF group, HPH-ADF group and CM-ADF group increased by 30.9%, 56.2% and 101.4%, respectively, and the CM-ADF group showed the most significant upward trend. These results indicate that CM-ADF significantly protected the activity level of GSH-PX in rat liver, possibly because ADF had a strong heavy metal adsorption capacity after compound modification, resulting in the ability to adsorb heavy metal Cd^2+^ in the intestinal tract, eliminate it from the body, and reduce the absorption of Cd^2+^ in rats, thus reducing the toxic effect of Cd^2+^ in the liver.

### 3.8. Protective Effect of ADF on Liver Tissue in Rats

Figure 5 shows the histological findings of the liver tissue of the BG, CG, HPH-ADF and CM-ADF groups of rats. BG group rats presented a normal structure; cell shading was relatively uniform, and the morphology of the liver tissue was normal. Blood vessels observed in the CG group indicated a considerable amount of extravasated blood (Figure 5A). This may have occurred because after Cd^2+^ was absorbed into the liver through blood vessels, the liver cells were damaged, and a large area of congestion was formed. The condition of multiple blood congestion was improved in the ADF, HPH-ADF and CM-ADF groups, although not reaching the standard of health, indicating that *artichoke* dietary fiber only plays a limited protective role and cannot replace drugs for treatment of this issue.

A considerable amount of hepatic cell steatosis was observed around the central vein and portal area of the liver, as well as in the liver parenchyma of CG group rats, and small white round vacuoles (blue circles) were observed in the cytoplasm (Figure 5B). There was almost no steatosis observed in the BG group. Steatosis in the liver cells of rats in the ADF, HPH-ADF and CM-ADF groups was gradually improved, with significant improvement in the CM-ADF group.

### 3.9. Protective Effect of ADF on Testicular Tissue in Rats

Figure 6 shows the histological findings of testicular tissue in rats. As shown in Figure 6A, the testicular cells of rats in the BG group were evenly arranged without edema, whereas the testicular tissue of rats in the CG group presented more severe edema (red circle). The testicular edema of rats in the ADF, HPH-ADF and CM-ADF groups was improved to some extent, with the most significant improvement observed in the CM-ADF group. The connective tissue in the testes of rats in the blank group was normally arranged, and the distance between spermatogenic tubules was normal (Figure 6B). In the CG group, the connective tissue arrangement in the testis was loose, the distance between spermatogenic tubules was widened and the separation from the boundary membrane was more serious (black arrow). Compared with the CG group, the ADF, HPH-ADF and CM-ADF groups showed significant improvement, with the most significant improvement in the CM-ADF group. Accumulation of Cd^2+^ in rats could lead to oxidative stress and autophagy, causing toxic effects on the body. Therefore, we speculate that *artichoke* DF modified by SP-HPH may reduce the toxic effects of Cd^2+^ by reducing oxidative stress and inhibiting autophagy.

## 4. Conclusions

In conclusion, after SP and HPH composite modification of ADF, the particle size of CM-ADF was reduced, with a uniform distribution; porous surface; and stronger oil-holding capacity, cation exchange capacity and total negative charge. CM-ADF had the strongest heavy metal adsorption capacity at both pH 2 and pH 7. We found that CM-ADF significantly reduced serum creatinine, urea, ALT, AST, white blood cell count and neutrophil ratio in the whole blood of cadmium-poisoned rats and increased liver GSH-PX activity, red blood cell count and hemoglobin count, in addition to reducing the effects of cadmium on the liver, kidneys and testes. In general, the microstructure, physical and chemical properties and health care function of ADF were improved by SP and HPH composite modification. This study provides reference data for the future modification of ADF.

## Figures and Tables

**Figure 1 foods-11-01716-f001:**
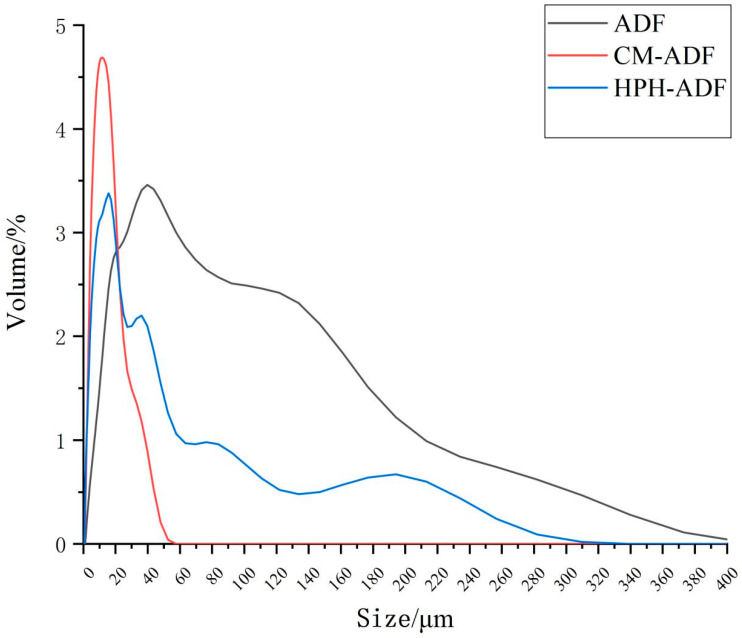
Particle size distribution of ADF, SP-ADF, HPH-ADF and CM-ADF.

**Figure 2 foods-11-01716-f002:**
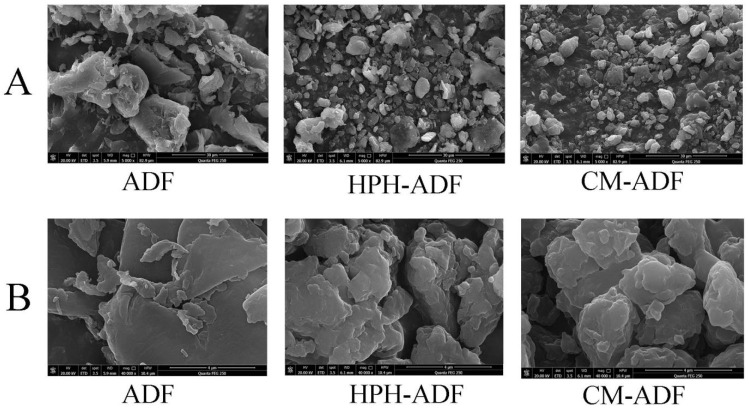
Scanning electron micrograph of ADF. (**A**) 5000 × magnification; (**B**) 40,000 × magnification. From left to right: ADF, HPH-ADF and CM-ADF.

**Figure 3 foods-11-01716-f003:**
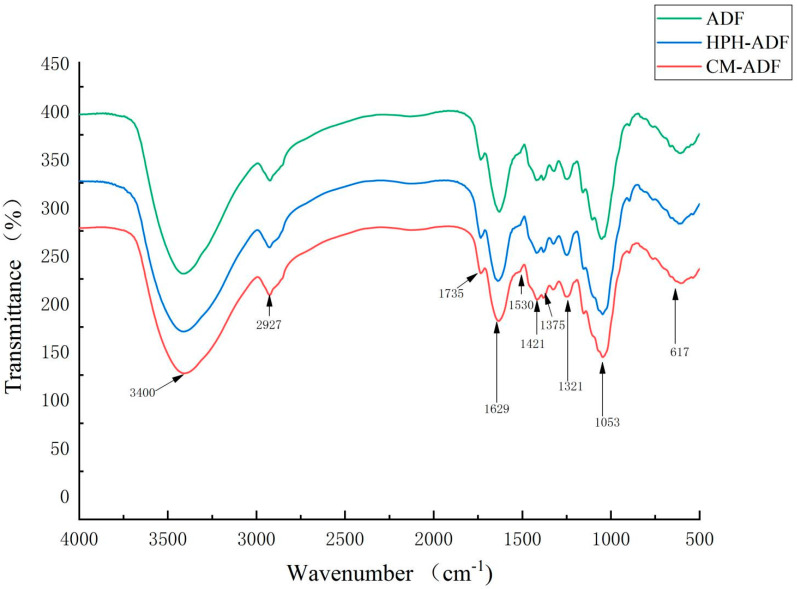
FTIR spectra of the ADF, HPH-ADF and CM-ADF groups.

**Figure 4 foods-11-01716-f004:**
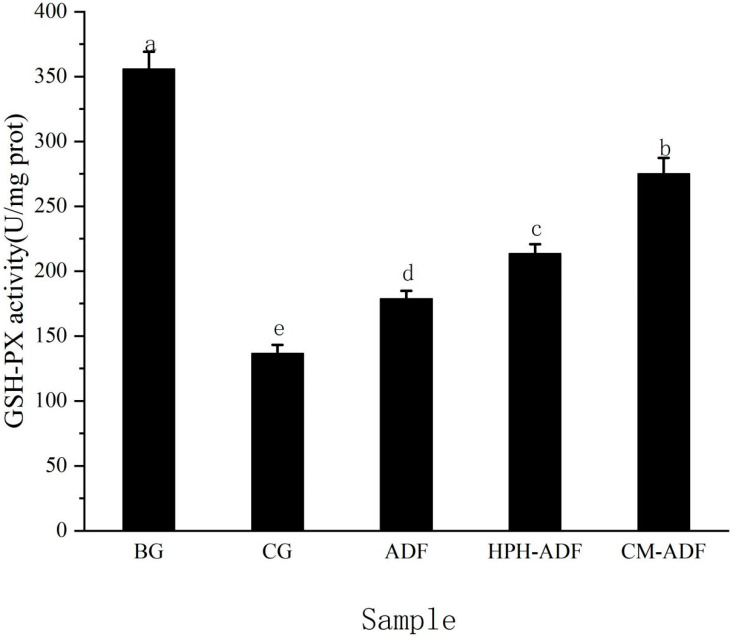
Effect of ADF on GSH-PX activity in rat liver. Data are reported as mean ± standard deviation from seven rats in each group, except for the blank group (five rats). Values with different letters are significantly different (*p* < 0.05).

**Figure 5 foods-11-01716-f005:**
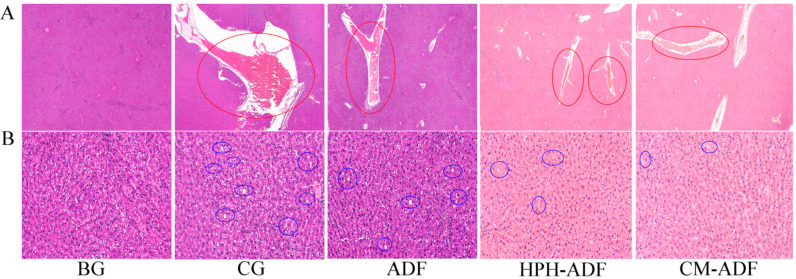
Hematoxylin-eosin-stained section of rat liver. (**A**) 2 × magnification; (**B**) 20 × magnification. From left to right: blank group (BG) with normal appearance, control group (CG), ADF group, HPH-ADF group and CM-ADF group. Hepatic congestion in red circles indicate hepatocyte damage, and round vacuoles in blue circles indicate hepatocyte steatosis.

**Figure 6 foods-11-01716-f006:**
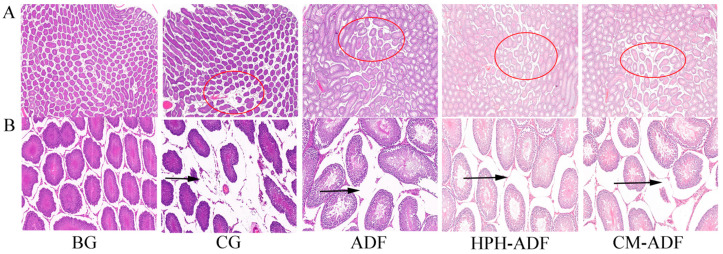
Hematoxylin-eosin-stained section of rat testis. (**A**) 2× magnification; (**B**) 20× magnification. From left to right: blank group (BG) showing normal appearance, control group (CG), ADF group, HPH-ADF group and CM-ADF group. Red circles indicate tissue edema; black arrows indicate significantly increased and decreased volume ratio of seminiferous tubules.

**Table 1 foods-11-01716-t001:** Experimental rat grouping and feeding.

Group	Number of Rats	Feeding	Drinking Water
BG	5	Basal diet	Deionized water
CG	7	Basal diet	50 mg/L aqueous solution of CdCl_2_
ADF	7	ADF special diet	50 mg/L aqueous solution of CdCl_2_
HPH-ADF	7	HPH-ADF special diet	50 mg/L aqueous solution of CdCl_2_
CM-ADF	7	CM-ADF special diet	50 mg/L aqueous solution of CdCl_2_

**Table 2 foods-11-01716-t002:** Effect of modification on the content of ADF.

Sample	SDF (g/100 g)	IDF (g/100 g)	TDF (g/100 g)
ADF	6.45 ± 1.26 ^d^	79.7 ± 1.36 ^a^	86.2 ± 1.33 ^b^
HPH-ADF	15.1 ± 1.38 ^b^	71 ± 0.98 ^c^	85.9 ± 1.25 ^d^
CM-ADF	21 ± 0.56 ^a^	66 ± 0.89 ^d^	86 ± 0.77 ^a^

Data are expressed as mean ± standard deviation (n = 3). Values with different letters in the same column are significantly different (*p* < 0.05).

**Table 3 foods-11-01716-t003:** Effect of modification on the WHC, OHC, TNC and SDF content of ADF.

Sample	WHC (g/g)	OHC (g/g)	TNC (mmol/g)	SDF Content (%)
ADF	5.8 ± 0.046 ^a^	2.3 ± 0.14 ^a^	2 ± 0.08 ^a^	7.6 ± 0.24 ^c^
HPH-ADF	3.6 ± 0.15 ^b^	2.8 ± 0.0.19 ^bc^	2.9 ± 0.15 ^b^	11 ± 0.32 ^b^
CM-ADF	2.9 ± 0.14 ^c^	3.2 ± 0.16 ^c^	5.3 ± 0.13 ^c^	14 ± 0.32 ^a^

Data are expressed by mean ± standard deviation (n = 3). Values with different letters in the same column are significantly different (*p* < 0.05).

**Table 4 foods-11-01716-t004:** ADF adsorption capacity of heavy metal ions.

Sample	Cd (II)/(mmol/g)		Cu (II)/(mmol/g)		Pb (II)/(mmol/g)		Hg (II)/(mmol/g)	
	pH2	pH7	pH2	pH7	pH2	pH7	pH2	pH7
ADF	3.5 ± 0.19 ^d^	12 ± 0.62 ^d^	2.9 ± 0.72 ^d^	12 ± 0.09 ^d^	12 ± 0.62 ^c^	24 ± 0.14 ^d^	3.1 ± 0.17 ^c^	10 ± 0.07 ^d^
HPH-ADF	4.0 ± 0.16 ^b^	24 ± 0.35 ^b^	6.9 ± 0.18 ^b^	19 ± 0.33 ^b^	13 ± 0.44 ^b^	41 ± 0.42 ^b^	4.1 ± 0.38 ^b^	21 ± 0.59 ^b^
CM-ADF	4.6 ± 0.26 ^a^	30 ± 0.46 ^a^	7.6 ± 0.43 ^a^	25 ± 0.13 ^a^	13 ± 0.37 ^a^	44 ± 0.22 ^a^	4.7 ± 0.24 ^a^	27 ± 0.12 ^a^

Data are expressed as mean ± standard deviation (n = 3). Values with different letters in the same column are significantly different (*p* < 0.05).

**Table 5 foods-11-01716-t005:** Serum AST, ALT, creatinine and urea levels in rats.

Group	AST (U/L)	ALT (U/L)	Creatinine (μmol/L)	Urea (mmol/L)
BG	42.6 ± 2.11 ^a^	114.4 ± 5.37 ^a^	37.2 ± 3.42 ^a^	5.3 ± 0.25 ^a^
CG	122 ± 5.84 ^e^	290 ± 9.20 ^e^	92.4 ± 4.93 ^e^	12 ± 0.52 ^e^
ADF	105 ± 2.71 ^d^	263 ± 6.79 ^d^	79.0 ± 4.04 ^d^	10 ± 0.47 ^d^
HPH-ADF	84.0 ± 2.27 ^c^	220 ± 10.8 ^c^	66.6 ± 5.22 ^c^	8.2 ± 0.39 ^c^
CM-ADF	73.0 ± 6.89 ^b^	152 ± 7.83 ^b^	54.43 ± 4.16 ^b^	6.5 ± 0.29 ^b^

Data are reported as mean ± standard deviation from seven rats in each group, except for the blank group (five rats). Values with different letters in the same column are significantly different (*p* < 0.05).

**Table 6 foods-11-01716-t006:** Blood routine indices in rats.

Group	Red Blood Cells (10^12^/L)	White Blood Cells (10^9^/L)	Neutrophil Proportion (%)	Hemoglobin Count (g/L)
BG	9.9 ± 0.95 ^a^	2.28 ± 1.02 ^a^	20.3 ± 7.22 ^a^	159 ± 8.84 ^b^
CG	7.94 ± 1.77 ^ab^	4.36 ± 1.79 ^ab^	29.2 ± 8.54 ^ab^	110 ± 3.53 ^a^
ADF	8.9 ± 0.77 ^ab^	3.99 ± 1.65 ^ab^	28.0 ± 6.36 ^ab^	113 ± 9.39 ^c^
HPH-ADF	8.8 ± 0.83 ^ab^	2.6 ± 0.93 ^bc^	27.0 ± 1.81 ^b^	119 ± 6.37 ^ab^
CM-ADF	8.79 ± 1.64 ^b^	2.6 ± 0.63 ^c^	25.6 ± 3.29 ^b^	152 ± 8.73 ^c^

Data are reported as mean ± standard deviation from seven rats in each group, except for the blank group (five rats). Values with different letters in the same column are significantly different (*p* < 0.05).

## Data Availability

Data is contained within the article.

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
