# Peer review of "Modification of Artichoke Dietary Fiber by Superfine Grinding and High-Pressure Homogenization and Its Protection against Cadmium Poisoning in Rats"

_foods, 2022, doi:10.3390/foods11121716_

Round 1

Reviewer 1 Report

This manuscript is well organized and written in acceptable manner. I have only minor comments given in the following list.

-        Line 36 and other places of the manuscript: all references in the manuscript are formally given in exponent. They have to be changed to the standard line (not up [1], but [1]).

-        Line 40 input gap between the” and “flowers”.

-        Line 49 input gap between “dietary” and “fibre”.

-        Lines 65-69 contain some health claims without any reference. Input corresponding reference that can confirm these claims.

-        Line 96 I think that there will be better to modify the sentence as “methods were explored to improve…”

-        Line 121 there is need to input the gap between “°C” and “and”.

-        Abbreviation list has to be enriched by SDF – soluble dietary fibre, used in Fig. 4.

Author Response

Dear Reviewer:

Reviewer 2 Report

The authors of this paper investigated the role of superfine grinding and high-pressure homogenization in modifying artichoke dietary fiber and its protection against Cadmium poisoning in rats. I strongly advise them to rework and resubmit their study after making the changes suggested in the following comments.

Generally, the abstract should be rewritten. It is not informative and does not reflect the aim of the research and the most important findings of the study. Please add p-value in the abstract when mentioned the word significant.

Abbreviations should be defined the first time they are mentioned.

The authors must identify the type of anesthetic used and the technique of blood collection in the materials and methods section.

 Red blood cell number, hemoglobin number, white blood cell number, and neutrophil ratio should be measured in whole blood and not serum, please check line 212 in the M&M section.

Data of food and water intake are not presented in the results section and the daily body weight of animals as well.

Although they claimed that this parameter had already been measured, no data on hemoglobin content was supplied.

It is written again the blood indexes were measured in serum (table 5 line 389), although, it must be in whole blood with heparinized samples.

Some markers (n=3), however, the number of animals 5-7 per group, please comment on this ??

What is the unit of GSH-PX ? the author should normalize the data of GSH-Px to the total protein content of each sample ( How much mg protein in each sample).

The hematoxylin-eosin stained section should be presented in high magnification to show the differences between treatments.       

Author Response

Dear Reviewer:

Reviewer 3 Report

This manuscript entitled “Modification of Artichoke Dietary Fiber by Superfine Grinding and High Pressure Homogenization and Its Protection on Cadmium Poisoning Rats” is an interesting and original study.

The paper is clearly presented and results are very useful. However, I have some suggestions:

1.     The results of total negative charge, oil holding capacity and water holding capacity due to include in only one table.

2.     Please when you write P<0.05 in the manuscript, replace it with p < 0.05

3.     Please, revise all tables mean and standard deviation: the number of digits of the mean value depends on the place where the significant digit appears and the number of digits of the corresponding data should be adjusted by taking into account the corresponding standard deviation values. In this way, each value in the Tables has been expressed with the significant digits according to the significant digits of each standard deviation value. Correct the different errors in the tables, keeping in mind that 0 and 1 are not significant digits.

Author Response

Dear Reviewer:

Round 2

Reviewer 2 Report

Despite the fact that the authors have responded to the majority of the comments, some have not.

The hematoxylin-eosin stained sections should be shown at high magnification with a detailed explanation of symbols such as arrows and circles on the pictures.

I strongly suggest the writers to unify the colors of the bar charts in all MS figures for each group.

Author Response

Dear reviewer:
